

# Effectiveness of Schroth exercises for adolescent idiopathic scoliosis: a meta-analysis

Yinfang Zhu, Caiying Zhu, Haiping Song and Manyan Zhang

Affiliated Hospital of Shaoxing University, Shaoxing, Zhejiang, China

## ABSTRACT

**Background:** Adolescent idiopathic scoliosis (AIS) affects 2–3% of children aged 10 to 18 and can lead to both physical and psychological challenges. This review aims to evaluate the impact of Schroth exercises on Cobb angle, angle of trunk rotation, and quality of life in patients with AIS.

**Methods:** A comprehensive search was conducted across Scopus, PubMed, CINAHL, Chinese National Knowledge Infrastructure, Cochrane Central, and ClinicalTrials.gov databases, identifying 11 randomized controlled trials (RCTs) with a total of 446 participants. These studies, primarily conducted in Asia and North America, compared the effects of Schroth exercises with either no treatment or alternative physical exercise interventions (standard physiotherapy regimens or non-specific exercise-based rehabilitation programs). Data were pooled using a random-effects model, and the quality of evidence was assessed using the Grading of Recommendations, Assessment, Development, and Evaluations (GRADE) approach.

**Results:** Meta-analysis showed that Schroth exercises were associated with a significant reduction in Cobb angle (standardized mean difference (SMD): −0.54; 95% confidence interval (CI) [−0.81 to −0.27]; $p < 0.001$). However, no significant effect was observed on the angle of trunk rotation (SMD: −0.90; 95% CI [−2.45 to 0.65]; $p = 0.254$). A significant improvement in quality of life was also noted (SMD: 0.67; 95% CI [0.33–1.01]; $p < 0.001$). The quality of evidence ranged from moderate to low.

**Conclusions:** Although Schroth exercises were associated with a reduction in Cobb angle, the average change did not meet the conventional five-degree threshold considered clinically significant, and the analysis did not account for the natural progression of spinal curvature. Rather than asserting a definitive benefit, this review highlights significant methodological limitations in the current literature. PROSPERO registration number: CRD42024505289.

# INTRODUCTION

Adolescent idiopathic scoliosis (AIS) is the most common form of pediatric scoliosis, characterized by a three-dimensional deformity of the spine that occurs in youth between the ages of 10 and 18 without a known cause (*Menger & Sin, 2024*). The condition affects 2% to 3% of the pediatric population, manifesting in varying degrees of spinal curvature

Corresponding author
Manyan Zhang,
zjsxmy85854357@163.com

and rotation (*Menger & Sin, 2024*). In addition to physical deformity, AIS is associated with a potential psychological distress, physical discomfort, and, in severe cases, compromised pulmonary and cardiac function (*Weinstein, 2019*; *Menger & Sin, 2024*). The management of AIS, therefore, is not only aimed at halting the progression of spinal curvature but also at minimizing its impact on the patient's quality of life and health (*Le Berre et al., 2019*; *Grabala et al., 2019*).

Traditional AIS treatment methods range from observation and bracing to surgical interventions, depending on the severity and progression of the curve (*Babaee et al., 2023*). In recent years, non-surgical approaches have increasingly gained traction, particularly physiotherapeutic scoliosis-specific exercises (PSSE), which encompass a variety of individualized exercise protocols aimed at managing mild to moderate cases of AIS. Emerging evidence supports the effectiveness of PSSE not only in reducing radiographic parameters but also in enhancing functional outcomes and quality of life (*Marchese et al., 2024*; *Negrini et al., 2018*).

Among the PSSE approaches, the Schroth exercises is one of the earliest and most extensively studied, having been developed in Germany in the 1920s by Katharina Schroth (*Weiss, 2011*). However, it is important to acknowledge that the term "Schroth" has evolved considerably over time. Many therapists now integrate elements from various PSSE techniques while still referring to their approach as 'Schroth' (*Marchese, Ilhan & Pacey, 2023*). For the purpose of this review, we define "Schroth" as the collection of exercise protocols that are rooted in the original methodology developed by Katharina Schroth, while recognizing that there is no single, standardized "Schroth" exercises in current practice. The core principle of this approach is that scoliosis is a complex three-dimensional condition, not merely a lateral deviation of the spine (*Schreiber, Whibley & Somers, 2023*). Its tailored exercises are designed to de-rotate, elongate, and stabilize the spine, thereby reducing curvature and asymmetry (*Seleviciene et al., 2022*).

While the original Schroth exercises, developed in early 20th century by Katharina Schroth, laid the foundation by focusing on corrective breathing and postural adjustments to address the three-dimensional spinal deformity, contemporary adaptations have significantly evolved (*Marchese, Ilhan & Pacey, 2023*). Today's variations integrate advanced physiotherapeutic principles and modern technology, resulting in more individualized exercise regimens that are tailored to a patient's specific curvature pattern. Notably, many practitioners now incorporate biofeedback, digital imaging, and enhanced proprioceptive training techniques to refine treatment precision. These modifications not only improve the adherence and engagement of patients, but also aim to optimize both the structural and functional outcomes of the exercise program. Exercises are designed to correct postural imbalances, strengthen weak muscles, and promote a more symmetrical posture through proprioceptive and kinesthetic training (*Weiss, 2011*; *Seleviciene et al., 2022*; *Schreiber, Whibley & Somers, 2023*). Principles of the Schroth exercises also emphasize the importance of patient education and self-management, encouraging AIS

patients to actively participate in the treatment and manage their condition more effectively.

Recent research on physiotherapeutic scoliosis-specific exercises has not only focused on improvements in spinal curvature, but also on other critical aspects of adolescent idiopathic scoliosis, including spinal deformities and functional abilities. Emerging evidence supports the effectiveness of PSSE programs in reducing radiographic parameters, enhancing muscle endurance, functional outcomes and quality of life (*Marchese et al., 2024*). Notably, recent work by *Marchese et al. (2024)* has detailed the ScoliBalance method, and *Karavidas et al. (2024)* have outlined the PSSE-Schroth approach, both of which provide a contemporary and in-depth perspective on PSSE interventions that address previously unrecognized components of the treatment framework. Likewise, studies by *Kuru et al. (2015)* and *Mohamed & Yousef (2021)* have reported improvements in balance and daily functional abilities, suggesting that these interventions may offer multifaceted benefits beyond structural correction. *Marchese, Ilhan & Pacey (2023)* found that clinicians practicing under the "Schroth" label routinely blend multiple exercise approaches, seldom delivering the original method in isolation. Given this evolution and the resulting variability, persisting in the use of "Schroth" can mislead both families and practitioners; we therefore advocate adopting more precise, practice-reflective terminology in future publications.

These findings underscore the importance of our meta-analysis, which aims to synthesize evidence across several outcome measures—including Cobb angle, trunk rotation, and quality of life—to provide a more comprehensive understanding of the benefits of Schroth exercises in the conservative management of AIS.

Despite its widespread use and anecdotal success stories, the effectiveness of the Schroth exercises in managing AIS is still unclear due to the variability in study designs, sample sizes, and outcome measures across research studies (*HwangBo, 2016*; *Duangkaew et al., 2019*; *Fang et al., 2022*). Although previous research has demonstrated the potential benefits of Schroth exercises for AIS, highlighting improvements in spinal curvature and quality of life, there remain substantial gaps, particularly regarding variability in exercise protocols, inconsistent effects on trunk rotation, and the overall heterogeneity of study designs. This study differs from earlier research by synthesizing data from geographically diverse set of randomized controlled trials and by incorporating rigorous quality assessment using the Grading of Recommendations, Assessment, Development, and Evaluations (GRADE) framework. By evaluating both structural outcomes and patient-centered measures, our meta-analysis provides a more comprehensive understanding of the clinical utility of Schroth exercises. In light of the evolving modifications in Schroth exercises and current gaps in standardization and comprehensive quality assessment, this review was undertaken to rigorously evaluate the effectiveness of Schroth exercises in adolescents with idiopathic scoliosis, thereby offering novel insights that clarify its role in conservative management and highlight areas for future research.

## METHODS

**Study Eligibility:** Studies were eligible for inclusion if they met the following criteria:

**Design:** Only randomized controlled trials (RCTs) were included in this meta-analysis because RCTs represent the gold standard for evaluating intervention effectiveness by minimizing bias and confounding factors. Cohort studies, case series, and case reports were excluded due to their higher risk of bias and the inherent limitations in establishing causal relationships, which could compromise the reliability of the pooled effect estimates.

**Population:** Adolescents diagnosed with idiopathic scoliosis (typically aged between 10 and 18 years).

**Intervention:** Studies providing intervention as Schroth exercises that typically comprised of individualized three-dimensional corrective exercises focussing on postural correction, targeted breathing techniques, and muscle strengthening, with session duration and frequency varying from 6 weeks to 6 months. Studies combining Schroth exercises with standard care were also eligible, given that control groups received conventional management for AIS (*e.g.*, routine physiotherapy, bracing, or observation), while intervention groups received these standard treatments supplemented by Schroth exercise regimen.

**Outcomes:** Reporting at least one of the following outcomes: Cobb angle, angle of trunk rotation, or quality of life.

Studies were excluded if they were non-RCTs (*e.g.*, observational studies, case series), did not focus specifically on AIS, or failed to provide sufficient data for extraction of the key outcomes. Additionally, abstracts, unpublished manuscripts, and non-peer-reviewed articles were excluded to ensure the reliability of the synthesized evidence.

### Search strategy

The search was done in Scopus, PubMed, CINAHL, Chinese National Knowledge Infrastructure (CNKI), Cochrane Central Register of Controlled Trials (CENTRAL) and ClinicalTrials.gov databases for studies, published from 1964 till January 2024, without any publication language restrictions. Both primary and secondary investigator (YZ and CZ) executed the search strategy in parallel and the search was executed on February 1 2024. Medical subject heading (MeSH) terms were combined with free text terms for literature search. Terms such as "Cobb's Angle", "Schroth gymnastics", "Adolescent Idiopathic Scoliosis", "Randomized Controlled Trial", "Angle of trunk rotation" were used in numerous combinations in these databases. Bibliographies of the selected studies were manually searched for additional relevant publications. In cases of unclear or missing data, addition, authors of the included trials were contacted.

### Screening and selection of records

Two independent reviewers (YZ and CZ) separately conducted searches, reviewed titles and abstracts for potential inclusion, and acquired full-text versions of relevant articles. The subsequent phase involved a detailed examination of abstracts and full texts by both the primary and the secondary based on predefined inclusion criteria. Any differences were

resolved through consensus or by consulting a third reviewer (HS), who also ensured the review process's integrity.

## Data collection methodology

The lead researcher (YZ) performed data collection during March to April 2024, including date of data extraction, the title and authors of the study, details from the methods section (such as study design, participant demographics, and setting), specifics about the participants (including the number in each group, baseline and outcome measurements, and criteria for inclusion and exclusion), details of the intervention and control groups, duration of follow-up, and outcomes (both primary and secondary measures, the timing of assessments, and additional information critical for quality assessment). Data on outcome metrics were extracted by both the primary and secondary investigators (YZ and CZ) from the chosen studies. In cases where trials featured multiple relevant groups, only those relevant to the analysis were considered.

## Assessment of study quality

Two independent reviewers evaluated the methodological quality of each included study using the Cochrane Collaboration's Risk of Bias 2 (RoB-2) tool (*Sterne et al., 2019*). This tool examines five key domains:

- Randomization process,
- deviations from the intended interventions,
- missing outcome data,
- measurement of the outcome, and
- selection of the reported result.

  Each domain was rated as "low risk," "some concerns," or "high risk" of bias. Discrepancies between reviewers were resolved through discussion, and if necessary, consultation with a third reviewer was sought to achieve consensus. Based on the assessment, studies were categorized as having a 'low,' 'high,' or 'some concerns' level of bias.

## Statistical analysis

STATA software, version 14.2 was used for the analysis. Data were reported as standardized mean difference (SMD) with 95% confidence interval (CI), based on the mean, standard deviation (SD) observed in the intervention and the control groups. Effect size having $p$-value less than 0.05 was considered statistically significant.

  A random-effects model with the inverse variance method was used to calculate effect sizes (*Cumpston et al., 2019*). This approach was chosen to better accommodate potential variations between studies. Heterogeneity among the included studies was assessed using both qualitative and quantitative methods. The $I^2$ statistic was calculated to quantify the proportion of total variation across studies attributable to heterogeneity rather than chance, with values interpreted as follows: less than 30% indicating low heterogeneity, 30–60% moderate heterogeneity, and greater than 60% high heterogeneity. In addition, Cochran's Q test was performed to determine the statistical significance of heterogeneity

across studies. A random-effects model with the inverse variance method was employed to account for variability among study outcomes, and sensitivity analyses were conducted to evaluate the robustness of the overall effect estimates. This analysis examined the influence of each individual study on the aggregate findings, thereby verifying the stability and reliability of the meta-analysis outcomes.

Given the limited number of studies included in our review (fewer than 10), conventional methods for detecting publication bias, such as Egger's test and funnel plots, were not used, and Doi plot and the Luis Furuya Kanamori (LFK) index were utilized instead. The LFK index, which ranges from −1 to +1, serves as an indicator of publication bias, with values outside this range suggesting varying degrees of asymmetry and potential bias (*Furuya-Kanamori, Barendregt & Doi, 2018*).

The assessment of evidence quality was guided by the Grading of Recommendations, Assessment, Development, and Evaluations (GRADE) framework (*Granholm, Alhazzani & Møller, 2019*), which systematically evaluates evidence across multiple dimensions: risk of bias (using the Cochrane risk of bias tools for this assessment), inconsistency (heterogeneity that was analysed using $I^2$ statistic and Cochran's Q test, to identify any significant variations in effect sizes), indirectness (how directly the evidence addressed the research question, including considerations of how applicable the study populations, interventions, and outcomes were to the specific research context), imprecision (confidence intervals surrounding the effect estimates were analysed to gauge the certainty of the conclusions drawn), and publication bias (LFK index, to detect any asymmetry that might suggest the omission of studies or the presence of small study effects).

Following this comprehensive evaluation, the quality of evidence was categorized into four levels: high, moderate, low, or very low to reflect the degree of confidence in the effect estimates.

## RESULTS

### Search results

Electronic search across the databases identified a total of 1,891 records. From this initial pool, 323 records were removed as duplicates, and additional 1,499 records were eliminated at the stage of title and abstract screening. Subsequently, 69 full texts were assessed for eligibility, and finally, 11 studies that met all the inclusion criteria were selected for the analysis (Fig. 1) (*Schreiber et al., 2015*, *2016*; *Kuru et al., 2015*; *Kim & HwangBo, 2016*; *HwangBo, 2016*; *Duangkaew et al., 2019*; *Lee & Lee, 2020*; *Kocaman et al., 2021*; *Gao et al., 2021*; *Mohamed & Yousef, 2021*; *Fang et al., 2022*).

### Characteristics of the included studies

This review includes 11 RCTs from diverse geographical regions, investigating the effects of Schroth exercises on patients with AIS. The studies originated from Korea (three), China (two), Turkey (two), Canada (two), Egypt (one), and Thailand (one), indicating a wide geographical spread. Sample sizes across these studies ranged from 15 to 863 participants, with intervention durations varying from 6 weeks to 2 years. Most of the studies (seven out of 11) reported a high risk of bias, while the remainder were classified as

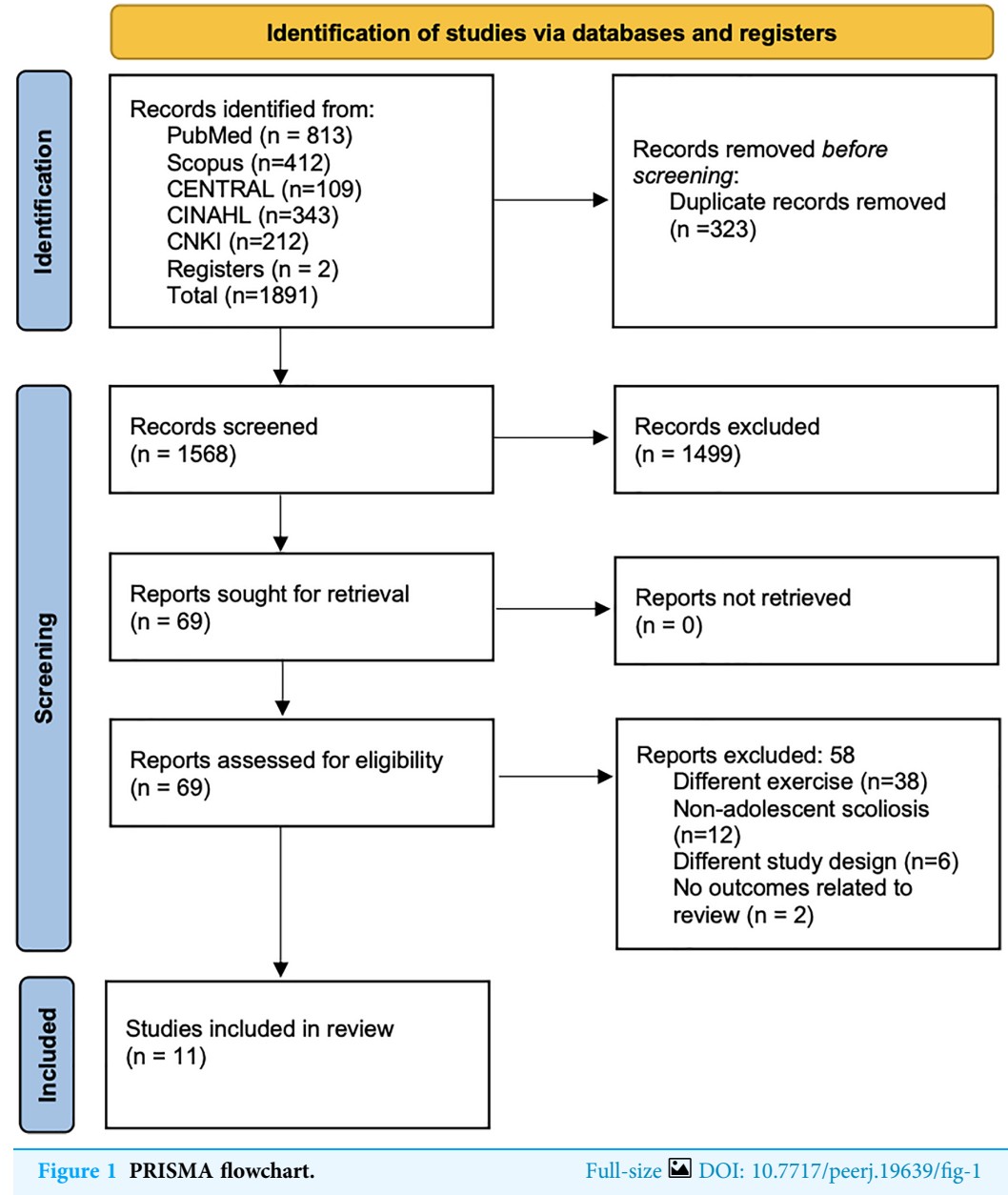

**Figure 1  PRISMA flowchart.**

having "some concerns" regarding bias (Table 1). The detailed different protocol and specific Schroth schools are provided in Table S1.

## Cobb's angle

The meta-analysis demonstrated a statistically significant reduction in Cobb angle with a standardized mean difference (SMD) of −0.54 (95% CI [−0.81 to −0.27]; $p < 0.001$) (Fig. 2), corresponding to a weighted mean difference (WMD) of −3.204 degrees. It is important to note that while this reduction is statistically significant, the magnitude of change is modest.

**Table 1 Characteristics of the included studies (N = 11).**

| Study identifier | Design | Study region | Participant details | Sample size | Intervention type | Duration of intervention | Outcomes assessed | Mean ± SD age in years | Risk of bias |
|---|---|---|---|---|---|---|---|---|---|
| Kim & HwangBo (2016) | RCT | Korea | Female students with scoliosis. | 24 | I-Schroth three-dimensional exercise C-Pilates exercise | 12 weeks | Cobb angle weight distribution | I = 15.6 ± 1.1 C = 15.3 ± 0.8 | High |
| Gao et al. (2021) | RCT | China | Patients diagnosed with AIS in Peking University Third Hospital. | 64 | I-Schroth three-dimensional exercise C-Observational group (no treatment) | 2 Years | Cobb angle sagittal vertical axis Coracoid height difference Thoracic kyphosis Thoracolumbar kyphosis | I = 15.1 ± 1.6 C = 15.8 ± 1.5 | High |
| Kocaman et al. (2021) | RCT | Turkey | Adolescents diagnosed with AIS by a physician according to Lenke criteria and referred to receive an exercise treatment at the School of Physical Therapy and Rehabilitation of Kırşehir Ahi Evran University | 28 | I-Schroth three-dimensional exercise C-Core stabilization exercise | 10 weeks | Cobb angle Trunk rotation angle (ATR) | I = 14.07 ± 2.37 C = 14.21 ± 2.19 | Some concerns |
| Kuru et al. (2015) | RCT | Turkey | Adolescent idiopathic scoliosis patients who applied to the Division of Physiotherapy and Rehabilitation, Faculty of Health Sciences, Istanbul University | 51 | I-Schroth three-dimensional exercise C-Observational group (no treatment) | 6 weeks | Cobb angle Trunk rotation angle (ATR) Quality of life | I = 12.9 ± 1.4 C = 12.8 ± 1.2 | Some concerns |
| Mohamed & Yousef (2021) | RCT | Egypt | Female students with right single thoracolumbar AIS, ranging in age from 14 to 16 years. | 34 | I-Schroth three-dimensional exercise C- proprioceptive Neuromuscular Facilitation technique (PNF) | 6 months | Cobb angle Trunk rotation angle (ATR) | I = 14.5 ± 1.20 C = 14.9 ± 1.4 | High |
| Fang et al. (2022) | RCT | China | Patients diagnosed with AIS. | 863 | I-Schroth three-dimensional exercise + Brace C-Brace | 6 weeks | Cobb angle Thoracic kyphosis Lumbar lordosis Quality of life | I = 12.5 ± 1.60 C = 12.29 ± 1.29 | High |
| Schreiber et al. (2016) | RCT | Canada | Patients with AIS aged 10–18 years, with curves of 10°–45° and Risser grade 0–5 from a single pediatric scoliosis clinic. | 50 | I-Schroth + standard of care C-standard of care | 6 months | Cobb angle Sum of curve | I = 13.5 ± 0.70 C =13.3 ± 0.5 | Some concern |

| Study identifier | Design | Study region | Participant details | Sample size | Intervention type | Duration of intervention | Outcomes assessed | Mean ± SD age in years | Risk of bias |
|---|---|---|---|---|---|---|---|---|---|
| *Lee & Lee (2020)* | RCT | Korea | Outpatients with scoliosis of The Goden clinic, South Korea | 15 | I-Schroth three-dimensional exercise C-Observational group (no treatment) | 12 weeks | Cobb angle ATR Total static plantar pressure Correction rate | I = 18.88 ± 3.06 C = 24.14 ± 12.6 | High |
| *Duangkaew et al. (2019)* | RCT | Thailand | Female volunteers, aged 10–18 years and with an angle trunk rotation of more than seven degrees. | 16 | I-Schroth three-dimensional exercise C-Kinesio tape with Schroth exercises | 6 weeks | ATR Inspiratory muscles strength Expiratory muscles strength Muscle endurance of back | I = 14.67 ± 2.34 C = 17.00 ± 1.63 | High |
| *Schreiber et al. (2015)* | RCT | Canada | Patients with AIS from the scoliosis, 10–18 years old, both genders, curves 10–45°, Risser 0–5. | 50 | I-Schroth + standard of care C- standard of care | 6 months | Quality of life Back extensor strength | I = 13.5 ± 0.70 C = 13.3 ± 0.5 | Some concern |
| *HwangBo (2016)* | RCT | Korea | Female high school students with scoliosis characterized by a Cobb's angle of 20° or greater. | 16 | I-Schroth three-dimensional exercise C-Pilate's exercise | 12 weeks | Cobb angle psychological factors | I = 18.14 ± 1.6 C = 18.88 ± 1.55 | High |

**Note:**
C, Control; I, Intervention; NR, Not reported; NA, Not applicable; RCT, randomized controlled trial; SD, Standard deviation; USA, United States of America.

In clinical practice, a change of approximately five degrees is often cited as the threshold for clinical relevance (*Keenan et al., 2014*). However, even a smaller reduction may be meaningful when considered in the broader context of a non-invasive intervention, particularly when accompanied by improvements in quality of life and other functional outcomes. We acknowledge that the clinical significance of a 3.204-degree reduction should be interpreted with caution, and further research is warranted to determine the long-term impact of these changes on patient management and outcomes. The moderate heterogeneity observed ($I^2 = 33\%$) indicates that, despite some variability among study protocols, the overall effect remains consistent. The LFK index was −1.39 with minor asymmetry in Doi plot, indicating minor possibility of publication bias (Fig. S1). Sensitivity analyses further confirmed that no single study unduly influenced the pooled estimate, supporting the robustness of these findings (Fig. S2).

## Angle of trunk rotation

The analysis of the angle of trunk rotation was done with the data from five studies (127 patients). The pooled SMD was −0.90, with a 95% CI ranging from [−2.45 to 0.65], indicating no statistically significant overall effect ($z = -1.141$, $p = 0.254$) (Fig. 3). There was a substantial heterogeneity among the included studies, with the $I^2$ value of 92.8% and

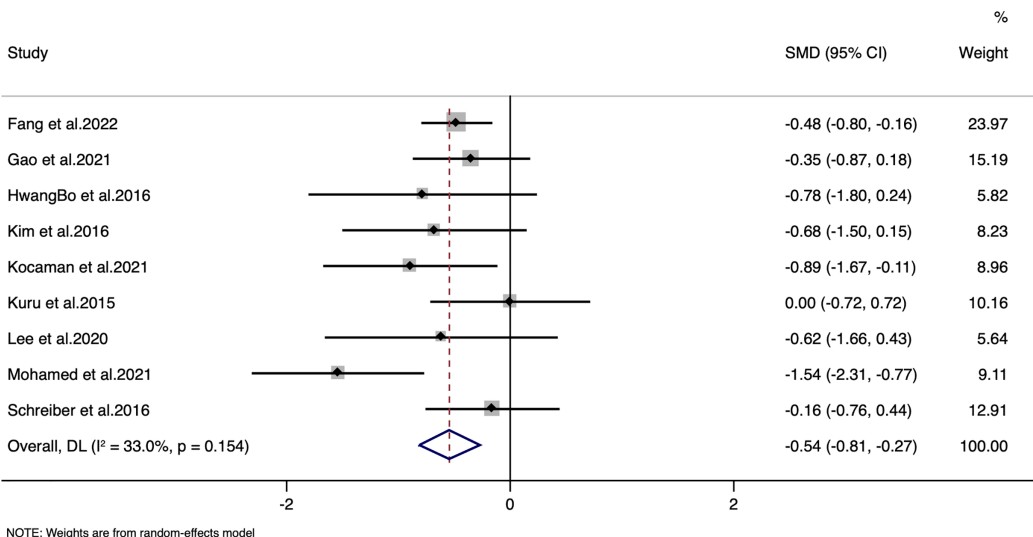

**Figure 2 Forest plot for Cobb's angle.**

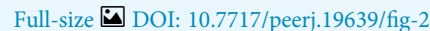
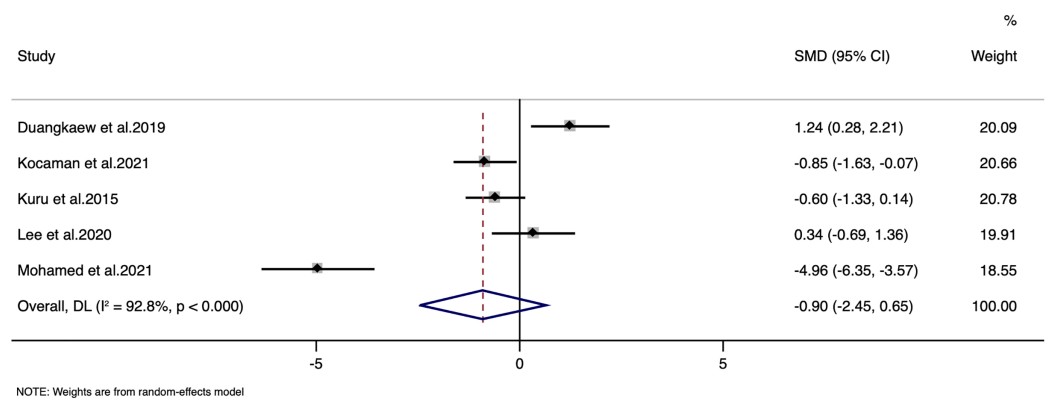

**Figure 3 Forest plot for angle of trunk rotation.**

a Cochran's Q statistic of 55.43 (df = 4, $p < 0.0001$). High heterogeneity ($I^2 = 92.8\%$) may arise from variations in exercise protocols, intervention durations, and measurement methods for trunk rotation. Additionally, differences in patient characteristics and adherence levels likely further contribute to this variability. The LFK index was −0.68 with no asymmetry in Doi plot, indicating no possibility of publication bias (Fig. S3). Sensitivity analysis indicate that there was no single study impact on the final pooled estimate (Fig. S4).

## Quality of life

Quality of life was reported in five studies with a combined total of 364 participants. Random-effects inverse-variance model analysis yielded a pooled SMD of 0.67, with a 95% CI ranging from 0.33 to 1.01 (Fig. 4). This result indicates a statistically significant overall

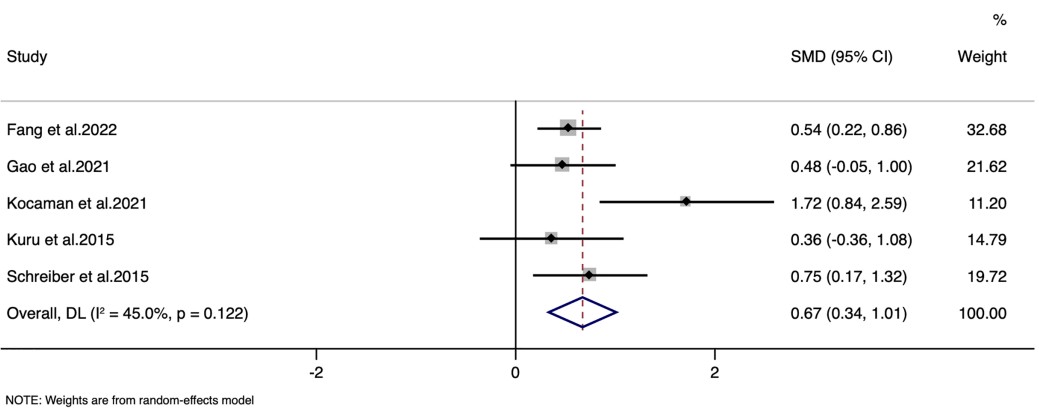

**Figure 4 Forest plot for quality of life.**

effect in improving the quality of life favouring the Schroth exercise intervention, with a z-value of 3.914 and a *p*-value < 0.001.

Heterogeneity among the included studies was moderate, with $I^2$ of 45.0% and a Cochran's Q statistic of 7.27 (df = 4), but this did not reach statistical significance (*p* = 0.122). The LFK index was 1.69 with minor asymmetry in Doi plot, indicating minor possibility of publication bias (Fig. S5). Sensitivity analysis indicate that there was no single study impact on the final pooled estimate (Fig. S6).

## GRADE assessment of evidence quality

The quality of evidence was appraised using the GRADE framework, which initially considers evidence from randomized controlled trials as high. However, several methodological shortcomings observed in the included studies necessitated downgrading the confidence in the effect estimates.

For Cobb angle, despite a consistent effect across studies, the quality was downgraded to moderate due to a high risk of bias. Many studies exhibited methodological weaknesses, including inadequate randomization procedures, lack of blinding of outcome assessors, and incomplete reporting of outcomes. Furthermore, moderate heterogeneity ($I^2$ = 33%) contributed to uncertainty regarding the true effect size.

For the angle of trunk rotation, the evidence was downgraded to low quality. This downgrading was primarily due to significant inconsistency among study results ($I^2$ = 92.8%), imprecision reflected in wide confidence intervals, and small sample sizes, all of which hinder confidence in the intervention's true effect. Additionally, variability in measurement techniques and reporting across studies further undermined the reliability of these findings.

Regarding quality of life, the evidence was rated as moderate, owing to concerns about minor publication bias and moderate heterogeneity ($I^2$ = 45%), despite a consistent direction of effect.

Overall, the methodological limitations—including non-standardized intervention protocols, insufficient blinding, and limited sample sizes—across multiple studies highlight

the need for more rigorously designed future trials to provide higher-quality evidence on the efficacy of Schroth exercises for the management of AIS.

## DISCUSSION

This review synthesized evidence from 11 randomized controlled trials, encompassing 446 participants, to evaluate the effects of Schroth exercises on patients with AIS. Although our meta-analysis indicates statistically significant reductions in Cobb angle and improvements in quality of life with Schroth exercises, these changes did not reach the conventional five-degree threshold typically associated with clinical significance. Moreover, due to the lack of data on the natural progression of scoliosis in the included studies, it is unclear whether the observed changes can be attributed solely to the intervention or reflect a pre-existing stabilization. Consequently, while Schroth exercises appear promising as a conservative management option for AIS, our findings primarily underscore the methodological limitations and heterogeneity in the current literature.

The analysis of the angle of trunk rotation revealed an insignificant overall effect (SMD: −0.90; 95% CI [−2.45 to 0.65]; $p = 0.254$) with substantial heterogeneity ($I^2 = 92.8\%$). Several factors may underlie these findings. First, the considerable heterogeneity suggests that differences in measurement techniques, exercise protocols, and patient characteristics across studies may have obscured a consistent effect. The limited number of studies and the small combined sample size (127 patients) further reduce the statistical power, potentially leading to an underestimation of the true effect. Moreover, the assessment of trunk rotation can be inherently challenging due to variability in measurement methods and potential for error, which might dilute the observed effect. Clinically, the insignificant impact on trunk rotation implies that while Schroth exercises appear effective in reducing structural deformity—as evidenced by improvements in Cobb angle—and in enhancing quality of life, they may be less effective in correcting the rotational component of AIS. This finding indicates that, in clinical practice, additional or complementary interventions might be required to specifically target and improve trunk rotation in patients with AIS.

Our findings on the efficacy of Schroth exercises in reducing Cobb angle and improving quality of life of AIS patients are consistent with prior research (*Seleviciene et al., 2022*; *Schreiber, Whibley & Somers, 2023*). Previous reviews have also reported improvements in spinal curvature and quality of life, supporting the adoption of Schroth exercises as a key component of conservative scoliosis treatment (*Burger et al., 2019*; *Dimitrijević et al., 2022*; *Ceballos-Laita et al., 2023*). However, the inconclusive results regarding the angle of trunk rotation diverge from some earlier studies that suggested a potential benefit, indicating the need for further research in this area.

Improvement in quality of life can be attributed to the targeted nature of Schroth exercises, which are designed to address the three-dimensional aspect of scoliosis. These exercises focus on de-rotating, elongating, and stabilizing the spine (*Weiss, 2011*; *Seleviciene et al., 2022*; *Schreiber, Whibley & Somers, 2023*). Moreover, the emphasis on patient education and self-management likely plays a crucial role in enhancing quality of life by empowering patients to actively participate in their treatment and manage their condition more effectively. While our meta-analysis has demonstrated significant

reduction in Cobb angle through Schroth exercises, the clinical implications of these findings warrant further exploration. Clinical significance, particularly in the context of AIS, refers to changes in treatment outcomes that meaningfully improve patient health and quality of life. In AIS, reductions in Cobb's angle that can delay or negate the need for surgical intervention are of paramount importance. Our findings indicate a mean decrease in Cobb angle of 3.204 degrees among participants undergoing Schroth exercises. According to the Society on Scoliosis Orthopaedic and Rehabilitation Treatment (SOSORT) guidelines (*Negrini et al., 2018*), a reduction of this magnitude being clinically significant is questionable.

Previous studies consistently demonstrated the efficacy of Schroth exercises in mitigating the progression of scoliosis curvature and enhancing patient well-being (*Burger et al., 2019*; *Dimitrijević et al., 2022*; *Ceballos-Laita et al., 2023*). However, discrepancies in outcomes related to the angle of trunk rotation highlight the variability in exercise implementation and the methodological diversity across studies. Notably, earlier research often lacked uniformity in exercise protocol and patient adherence, which may contribute to the observed mixed results. This underscores the importance of standardizing Schroth exercise protocols in future research to ensure consistency and comparability of outcomes.

One of the strengths of this study is the comprehensive search strategy and inclusion of RCTs from diverse geographical regions, enhancing the generalizability of the findings. Additionally, application of the GRADE framework to assess the quality of evidence adds rigor to our conclusions. Our study has some limitations. The moderate to high risk of bias in most included studies and the moderate heterogeneity observed in our analyses may affect the reliability of the findings. Furthermore, limited number of studies addressing some outcomes, such as the angle of trunk rotation, underscores the need for cautious interpretation of these results. In addition to the aforementioned limitations, several other factors should be considered. First, the heterogeneity of intervention protocols—including differences in exercise regimens, patient adherence, and follow-up durations—may have influenced the pooled effect sizes and contributed to the moderate heterogeneity observed. Furthermore, although our search strategy was comprehensive, the possibility of language bias and the omission of unpublished studies cannot be entirely ruled out. Finally, the relatively short follow-up periods in most studies limit our ability to assess the long-term efficacy and safety of Schroth exercises.

In addition to addressing the current limitations, future research should provide more detailed information regarding the specific methods and intensity of the Schroth exercises implemented in each trial (*e.g.*, frequency, duration, and level of supervision). Such details would allow for a clearer understanding of whether variations in exercise protocols contribute to the heterogeneity observed in the outcomes. Moreover, it would be beneficial to explore the influence of patient characteristics—such as age, gender, and baseline scoliosis severity—on the effectiveness of the intervention. Incorporating a broader range of outcome measures, including assessments of psychological well-being and functional capacity, would also help capture the multidimensional benefits of Schroth exercises and provide a more comprehensive evaluation of their impact on quality of life. Future research should aim to address the existing gaps in the literature, particularly by conducting

high-quality RCTs with larger sample sizes and longer follow-up periods to evaluate the long-term efficacy of Schroth exercises. Additionally, exploring the impact of these exercises on other outcomes, such as trunk rotation and pulmonary function, would provide a more comprehensive understanding of their benefits. Incorporating standardized outcome measures and reporting practices would also facilitate more accurate comparisons across studies.

## CONCLUSION

The overall reduction in Cobb angle did not reach the conventional five-degree threshold commonly defined as clinically significant, and our analysis did not account for the rate of natural curve progression. While we observed statistically significant changes associated with the introduction of targeted corrective breathing and postural exercises for techniques originating from the Schroth approach, the magnitude of these improvements fell below commonly accepted thresholds for clinical relevance. Importantly, none of the included trials fully characterized therapist expertise, supervision intensity or patient-borne costs, all of which may influence outcomes. Moreover, as previous studies demonstrate, practitioners operating under the "Schroth" label frequently blend multiple modalities; future studies should therefore specify individual corrective techniques rather than rely on the umbrella term "Schroth". Consequently, it remains uncertain whether the observed changes reflect a true treatment effect or simply a manifestation of spontaneous stabilization, particularly since baseline progression rates were not uniformly reported across studies. Rather than asserting a guaranteed benefit of Schroth exercises, our study highlights significant methodological limitations in the current literature. These findings underscore the need for future well-controlled, high-quality trials that not only use standardized exercise protocols and longer follow-up durations but also meticulously track the natural history of curve progression in AIS patients. Future research should specifically focus on evaluating the differential effects of Schroth exercises according to curve type (*e.g.*, thoracic, lumbar, and thoracolumbar) and among patients with varying skeletal maturity levels as indicated by Risser stages. Additionally, further work is required to delineate the optimal exercise protocols and dosage for different subpopulations to ensure more individualized, effective conservative treatment approach for adolescent idiopathic scoliosis.

### Funding
The authors received no funding for this work.

### Competing Interests
The authors declare that they have no competing interests.

### Author Contributions
- Yinfang Zhu conceived and designed the experiments, authored or reviewed drafts of the article, and approved the final draft.

- Caiying Zhu performed the experiments, analyzed the data, prepared figures and/or tables, authored or reviewed drafts of the article, and approved the final draft.
- Haiping Song performed the experiments, analyzed the data, prepared figures and/or tables, authored or reviewed drafts of the article, and approved the final draft.
- Manyan Zhang performed the experiments, prepared figures and/or tables, and approved the final draft.

### Data Availability

This is a systematic review/meta-analysis.

### Supplemental Information

Supplemental information for this article can be found online at http://dx.doi.org/10.7717/peerj.19639#supplemental-information.

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
