# Peer review of "Effectiveness of Schroth exercises for adolescent idiopathic scoliosis: a meta-analysis"

_PeerJ, doi:10.7717/peerj.19639_

## Round 0.1 · original submission · Major Revisions

· Academic Editor

Major Revisions

Reviewer 3 has recommended rejection, so you should pay particular attention to address their comments in a substantial way

·

Basic reporting

The manuscript provides clear and comprehensive basic reporting. It is written in well-structured and concise English, with an adequate introduction outlining the prevalence and challenges of Adolescent Idiopathic Scoliosis (AIS). Relevant background information and previous literature are cited to contextualize the study. The methods section describes a systematic and transparent approach to data collection and analysis, including the use of robust databases and the GRADE framework. Figures and results are well-presented, and the conclusions align with the data. However, additional elaboration on the limitations and the broader implications of the findings would further strengthen the reporting.

Experimental design

The experimental design of the studies included in this meta-analysis is clearly outlined. It involves randomized controlled trials (RCTs), which are considered the gold standard for assessing the effectiveness of interventions. The studies compared the effects of Schroth exercises to either no treatment or other forms of physical exercises, providing a robust comparison. The sample size of 446 participants across diverse geographical regions strengthens the external validity of the findings. However, while the design appears sound, the short follow-up periods in some studies are a limitation that affects the ability to assess long-term effects. Future studies should address this gap with extended follow-up durations to evaluate the lasting impact of the interventions.

Validity of the findings

The validity of the findings is generally strong, particularly given the use of randomized controlled trials (RCTs), which are considered the highest level of evidence for assessing the efficacy of an intervention. The meta-analysis included a substantial number of studies (11 RCTs with 446 participants), which enhances the statistical power and reliability of the results. The pooled data analysis using a random-effects model is an appropriate approach for synthesizing the results from diverse studies. However, the quality of evidence varied from moderate to low, as indicated by the Grading of Recommendations, Assessment, Development, and Evaluations (GRADE) approach. This variation in evidence quality suggests that while the findings are promising, further high-quality studies with longer follow-up periods are needed to confirm the long-term effectiveness and applicability of Schroth exercises for AIS patients. Additionally, the lack of significant effects on trunk rotation angle calls for more investigation to fully understand the scope of the exercises' impact.

Additional comments

In addition to the points mentioned, it would be beneficial for future research to include more detailed information on the specific methods and intensity of the Schroth exercises used in the trials. This would help to clarify whether the variations in exercise protocols across studies contributed to the heterogeneity of the results. Moreover, it would be valuable to explore how factors such as age, gender, and the severity of scoliosis might influence the effectiveness of the exercises. Future studies could also consider including a wider range of outcome measures to assess other aspects of quality of life, such as psychological well-being and functional capacity, which could provide a more comprehensive understanding of the benefits of Schroth exercises. Lastly, addressing the limitation of short follow-up periods by incorporating longer-term studies would provide insights into the sustainability of the improvements observed with Schroth exercises.

Reviewer 2 ·

Basic reporting

1. Language and Clarity:
The manuscript is written in clear and professional English. The terminology used is appropriate for the intended academic audience, and technical details are articulated unambiguously. However, minor grammatical and syntactic improvements could enhance readability, particularly in the introduction and discussion sections.
2. Professional Standards:
The manuscript adheres to professional standards of courtesy and academic expression. The authors maintain an objective tone throughout.
3. Introduction and Background:
The introduction provides a comprehensive background, establishing the relevance of Schroth exercises in the treatment of adolescent idiopathic scoliosis (AIS). Relevant prior literature is cited, demonstrating the research's connection to the broader field. However, the introduction could benefit from a more detailed discussion of gaps in existing literature and the specific novelty of this study.
4. Structure and Formatting:
The manuscript follows the standard scientific structure (Abstract, Introduction, Methods, Results, Discussion, and Conclusion). Each section flows logically, ensuring clarity. The adherence to PeerJ guidelines and discipline norms is evident.
5. Figures and Tables:
Figures are relevant and contribute to the manuscript's clarity. They are well-labeled and appropriately described. The resolution is sufficient for readability. However, Figure 3 (Angle of Trunk Rotation) could benefit from additional annotation to enhance clarity.
6. Data Availability:
The manuscript follows data-sharing policies, making appropriate raw data available. The supplementary materials are well-organized and contribute to the transparency of the analysis.

Experimental design

1. Scope and Relevance:
The study falls within the journal's scope, focusing on the clinically significant topic of AIS treatment through Schroth gymnastics.
2. Research Question:
The research question is well-defined, relevant, and meaningful. The manuscript clearly states how the study addresses an identified knowledge gap, specifically in evaluating Schroth exercises' effectiveness on Cobb’s angle, trunk rotation, and quality of life.
3. Methodology:
The study employs rigorous methods, including a systematic literature review and meta-analysis of randomized controlled trials (RCTs). The methodology is described in sufficient detail to allow replication, covering search strategy, inclusion criteria, and statistical analysis.
4. Ethical Considerations:
The study does not require ethical approval as it relies on secondary data analysis. The authors have appropriately acknowledged this in the manuscript.

Validity of the findings

1. Data Robustness and Statistical Analysis:
The statistical methods used (random-effects model, sensitivity analysis, GRADE assessment) are appropriate and robust. The results are presented with clear confidence intervals and significance values. However, the discussion could better contextualize the clinical significance of the findings.
2. Conclusions and Limitations:
The conclusions are well-linked to the research question and results. The authors acknowledge the limitations, such as short follow-up periods and heterogeneity among included studies. These acknowledgments enhance the study's credibility.
3. Reproducibility:
The findings appear reproducible given the transparency of data reporting. However, potential biases due to differences in intervention protocols among studies should be discussed further.

Additional comments

Strengths:
• Comprehensive meta-analysis with a robust methodological framework.
• Clear articulation of the relevance and impact of Schroth exercises on AIS.
• Use of appropriate statistical techniques to assess heterogeneity and bias.
Weaknesses and Suggestions for Improvement:
• Introduction: Expand on the uniqueness of the study compared to previous meta-analyses.
• Methods: A more detailed discussion on the exclusion criteria and potential selection biases would strengthen the methodology section.
• Discussion: While the authors mention clinical significance, a deeper interpretation of how the reduction in Cobb’s angle translates into practical clinical outcomes would be valuable.
• Figures: Consider improving clarity in Figure 3 to better convey results related to trunk rotation.

Reviewer 3 ·

Basic reporting

The title should not include the use of the word 'gymnastics' in association with Schroth as it is misleading. The authors seem to be addressing Schroth and in the general population 'gymnastics' means something different worldwide.

The English needs to be checked, e.g. 'Cobb’s angle' should be 'Cobb angle'

The authors should acknowledge PSSE (physiotherapeutic scoliosis specific exercises' as a whole before introducing 'Schroth' which is considered one type of PSSE. For example, this sentence 'Traditional AIS treatment methods are ranging from observation and bracing to surgical
49 interventions, depending on the severity and progression of the curve (Babaee et al., 2023)' does not even mention PSSE. It would be appropriate, given the nature of this article, that the authors more appropriately introduce PSSE, the emerging evidence behind it, and how 'Schroth' fits into this. Also the definition of 'Schroth' being so varied worldwide should be acknowledged because there really is no true 'Schroth' method anymore. 'Schroth' is Katharina's last name and the method has evolved into many methods with many therapists now mixing their methods. A proper definition as to what they are referring to as 'Schroth' should be included. When the authors do introduce it, the reference is inappropriate. 'However, in recent years, non-surgical approaches, particularly physiotherapeutic scoliosis-
51 specific exercises (PSSE), have gained popularity due to their effectiveness in managing mild to
52 moderate cases of AIS (Berdishevsky et al., 2016).' Berdishevky's paper is simply a review of some PSSE methods and is now outdated with many more PSSE methods having evolved. While the authors do attempt to describe it, and indeed refer to Schroth being a 'stand out' and 'individualised', this is biased and potentially deceiving. There are multiple other methods that individualise their programs. Line 66 references should also acknowledge that many Schroth therapists mix their methods even though they call themselves Schroth therapists (Marchese et al, 2023).

Paragraph on line 265 and 279 is repeated.

Experimental design

I cannot comment on the statistics but I am not convinced the papers justify a meta-analysis but only a systematic review. A more in-depth explanation of the exercises in the papers used would have been appropriate.

Validity of the findings

I cannot comment on the statistics however in one comment they say the results are significant but then the authors say the reduction in Cobb was 'Our findings indicate a mean decrease in Cobb’s angle of 3.204 degrees among participants 239 undergoing Schroth exercises. This is not clinically significant. It is also a very misleading statement.

The authors do not examine whether the RCTs vary in dosage and description and whether the methods used in each paper were actually truly Schroth. Many Schroth therapists use a lot of Corrective Breathing which is arduous on the patient and arduous on the therapist to get results. How much effort was applied to acheive only a 3degree reduction. I would have also liked to have seen the authors look into any mention of TRACE scores so that the patient aesthetics were taken into account. Although TRACE has poor inter-rater reliability the intra-rater reliability is good and most, if not all, of the papers used in this review do not use TRACE. A comment on this might be appropriate given that many patients go to Schroth therapists for aesthetic reasons not just Cobb angle.

The conclusion is misleading - ' Our meta-analysis indicates that Schroth exercises are effective in reducing Cobb’s angle and
309 improving quality of life in patients with AIS. However, due to the short follow-up periods in our
310 study, these conclusions must be viewed cautiously. The impact on the angle of trunk rotation
311 and long-term outcomes remains unclear, necessitating further research with longer follow-up to
312 fully understand the benefits.'

A change of more than 5 degrees needs to occur for the change in Cobb angle to be meaningful and to account for human error. The authors acknowledge that a change this small is not that great, however perhaps it may have been worthwhile to examine that if you are dealing with a proven highly progressive curve, with Risser signs at similar stages, that stabilisation of the curve is achieveable with Schroth.

Additional comments

This paper is misleading. In one instance the authors acknolwedge that the Cobb angle change is not that large and then the conclusion says otherwise. In the context of stabilisation of highly progressive curves it may be appropriate but that was not made clear.

---

## Round 0.2 · Major Revisions

· Academic Editor

Major Revisions

Reviewer 2 still has some major concerns about the manuscript.

·

Basic reporting

Abstract
The abstract clearly demonstrates the significance of the study on the impact of Schroth exercises on AIS, with a well-structured methodological approach and precise presentation of key results. It is particularly commendable that relevant databases were covered and the GRADE methodology was employed to assess the quality of evidence.
However, it is recommended to improve the precision in defining the comparator (other types of physical exercises) and the geographic regions from which the studies originate, to facilitate a clearer understanding of the methodological approach.

Introduction
The introduction is well-structured and provides a clear overview of adolescent idiopathic scoliosis (AIS), its significance, and various therapeutic approaches. The author successfully contextualized the problem, highlighted the relevance of the Schroth method, and clearly defined the study’s objectives. It is praiseworthy that numerous relevant sources are referenced and modern citations are used to support claims.

Possible improvements:

Experimental design

Introduction
Precision in Terminology:
Clarify how today’s variations of the Schroth method differ from the original methodology. For example, include a note on the modifications introduced over the years.
Consistency in Terminology:
Use a consistent term throughout the manuscript (e.g., use only "Schroth exercises" instead of "Schroth gymnastics training" or "Schroth method").
Conclusion of the Introduction:
Better link the final sentence to the preceding text for a more natural conclusion of the introduction. For example, emphasize how this study differs from previous research and what it specifically contributes that is new compared to earlier studies.
Methods
Precision in Defining the Intervention:
Provide more details about the specific exercises or protocols included in the Schroth training, if such data are available in the selected studies. Clarify how combinations with standard care were defined in the analysis.
Consistency in Terminology:
Standardize the terminology and use a consistent term throughout the manuscript (e.g., use only "Schroth exercises" instead of "Schroth gymnastics" or "Schroth training").
Justification for Excluding Certain Types of Studies:
Briefly explain why certain types of studies (e.g., cohort studies or case studies) were excluded from the analysis and the rationale behind their exclusion.
Results
Clarify the Clinical Significance Threshold for Cobb's Angle:
Provide a reference for the claim that a 5-degree change is commonly used as the clinical significance threshold, or further discuss its relevance in clinical practice.
Discuss High Heterogeneity in Trunk Rotation Angles:
Discuss possible sources of variability among studies, such as differences in exercise protocols, intervention durations, and measurement methods, given the high heterogeneity (I² = 92.8%).
Provide Visual Representations of Key Findings:
Include graphs or tables with data on the effects of the intervention on Cobb’s angle, trunk rotation angle, and quality of life to improve the clarity and comprehensibility of the results.
Consider Limitations in Interpreting Quality of Life Results:
Consider the use of validated questionnaires for measuring quality of life and discuss their reliability in the context of the studies analyzed.
Conclusion
Clearly Define the Need for Further Research:
Instead of a general statement about the need for longer follow-up, specify which particular knowledge gaps should be addressed (e.g., effects of Schroth exercises on different types of curves or patients with varying Risser stages).

Validity of the findings

no comment

Reviewer 3 ·

Basic reporting

1. I continue to believe that the conclusions are misleading. ' Our meta-analysis indicates that Schroth exercises are associated with a statistically significant reduction in Cobb angle and improvements in quality of life in patients with AIS Although this reduction does not reach conventional 5-degree threshold typically required for clinical significance, it may still reflect stabilizing effect in highly progressive curves—particularly among patients with similar Risser staging.

This statement leads with the most positive aspect of the results, which, in this instance, is actually misleading.

2. This comment is false - The Schroth method stands out for its individualized approach.

Many PSSE methods take an individualised approach these days.

3. References are old/outdated. Many new PSSE methods now exist. Using these references, Berdishevsky et al. (2016) and Burger et al. (2019) are false and misleading. Marchese et al have published a paper outlining in clear detail the ScoliBalance method, and Karavidas has published papers outlining the PSSE-Schroth method. These papers are all more recent. Berdishevsky's paper is 9 years old and missing critical pieces of the PSSE world.

4. 'Schroth gymnastics training' should be removed. This is misleading. There is still confusion in this paper as to what constitutes Schroth, but either way, the conclusion is skewed towards favorable reporting and is misleading.

5. It seems no change was found, and without knowing the rate of progression of each of the curves, even stabilisation claims are false. It is possible that the curves were stabilising before treatment. It is imperative that we get good research in this space, however, I think this paper has the potential to better point out the flaws in many of the Schroth papers to drive better research rather than try to manipulate the results to indicate that Schroth results are guaranteed.

Experimental design

No comment

Validity of the findings

As above

Additional comments

This paper is skewing towards trying to find a way to say that Schroth is beneficial. While clinically it has been seen that this occurs, none of the published papers show strong evidence of this. The conclusion is misleading.

I think this paper has great potential. But the authors should relook at the data and the reporting of the conclusions being made. Essentially, not much has been proven yet from many of the papers. None of them has the Schroth protocol outlined properly.

---

## Round 0.3 · Minor Revisions

· Academic Editor

Minor Revisions

Reviewer 3 ·

Basic reporting

1. References of Berdishevsky et al are not accurate. This paper does not show the benefits of PSSE but rather reports on the types of schools. It is now outdated as there are multiple other schools. Also Marchese et al, 2024 showed that muscle endurance can be increased with PSSE (ScoliBalance) and bracing combined.

2. Line 55 spelling mistake - should be 'considerably'

3. 'Many therapists now integrate elements from various PSSE techniques while still referring
57 to their approach as ‘Schroth’ (Seleviciene et al., 2022).' The better referrence for this is that Marchese et al 2023 showed that Schroth therapists mix their methods and actually don't do any pure Schroth. The Schroth method now has evolved and we really should move away from the use of the word Schroth in papers as it confusing for parents and health professionals.

4. Line 66 should be referenced Marchese et al, 2023.

5. Line 80 does not even mention muscle endurance, covered by Marchese et al, 2024.

6. Noteably, lines 82 -84 starts to recognise some of the newer methods but it needs to be combined with the other items above.

Experimental design

Good

Validity of the findings

Line 241 'These results underscore the potential value of integrating Schroth exercises into
242 conservative management plans for AIS patients, providing a non-invasive strategy that can positively impact both radiographic outcomes and patient-reported measures.' This statement this is misleading. A statistically significant results was found that was not clinically significant. While it is possible that Schroth may have some effect on the patient results we don't know if it is clinically significant in the context of the studies we have. Not enough emphasis on the experience of the Schroth therapist is being noted, how much supervision was required, expense on the patient etc. Schroth may have it's place but it's now well established that Schroth therapists mix their methods (Marchese et al, 2023) and thus this study may be better off addressing the concepts of Schroth not the use of the word Schroth, i.e. many methods now use Corrective breathing, which originated from Schroth, but is not purse Schroth itself.

Additional comments

The conclusion has improved. See comments above for some further clarity about the Cobb angle reduction that is not clinically significant.

---

## Round 0.4 · accepted · Accept

· Academic Editor

Accept

The authors are commended on their attention to detail and for addressing the reviewer comments. Your manuscript is now suitable for acceptance.

Reviewer 3 ·

Basic reporting

I thank the authors for the well-written adjustments to the paper. It more clearly explains the challenges of Schroth in the way results are reported.

Experimental design

Good

Validity of the findings

No comment

Additional comments

Typo 'Cobb's' should be 'Cobb'